# Active Learning for Multiple Target Models

**Ying-Peng Tang and Sheng-Jun Huang** *
College of Computer Science and Technology, Nanjing University of Aeronautics and Astronautics
Collaborative Innovation Center of Novel Software Technology and Industrialization
MIIT Key Laboratory of Pattern Analysis and Machine Intelligence, Nanjing 211106, China
{tangyp,huangsj}@nuaa.edu.cn

## Abstract

We describe and explore a novel setting of active learning (AL), where there are multiple target models to be learned simultaneously. In many real applications, the machine learning system is required to be deployed on diverse devices with varying computational resources (e.g., workstation, mobile phone, edge devices), which leads to the demand of training multiple target models on the same labeled dataset. However, it is generally believed that AL is model-dependent and untransferable, i.e., the data queried by one model may be less effective for training another model. This phenomenon naturally raises a question "*Does there exist an AL method that is effective for multiple target models?*". In this paper, we answer this question by theoretically analyzing the label complexity of active and passive learning under the setting with multiple target models, and conclude that AL does have potential to achieve better label complexity under this novel setting. Based on this insight, we further propose an agnostic AL sampling strategy to select the examples located in the joint disagreement regions of different target models. The experimental results on the OCR benchmarks show that the proposed method can significantly surpass the traditional active and passive learning methods under this challenging setting.

## 1 Introduction

Data labeling is expensive due to the involving of human annotator. Active learning (AL) is one of the main approaches to reduce the labeling cost [28]. It evaluates the utility of the unlabeled data based on the target model, and actively queries the labels from the oracle for the examples that is the most beneficial to the performance improvement of the target model.

Existing active learning methods assume that there is only one specific target model, and try to fit it with least queries. However, in many real applications, the machine learning system is required to be deployed on multiple types of devices with different resource constraints [6]. For example, a speech recognition software needs to support diverse machines with varying hardware efficiency, ranging from high-performance workstation to the mobile-phone. Due to the different computational resources, the applicable model architectures vary a lot, e.g., a deep model which is well-performed on the cloud server may not be deployed on the edge device. It thus raises the demand of training multiple models with different complexity to accommodate these devices.

Given multiple target models, how to effectively improve them with least labeled data becomes a practical and challenging problem. It is generally believed that AL is usually model-dependent and untransferable [22, 24, 38], i.e., the best query strategy for different target models varies a lot [39]. In other words, the data queried by one model may be less effective for training another model [22]. These observations imply that the existing active query strategies can hardly benefit all target models simultaneously, and the design of AL algorithm for multi-models can be rather difficult. A natural

---

*Corresponding author: Sheng-Jun Huang <huangsj@nuaa.edu.cn>.

36th Conference on Neural Information Processing Systems (NeurIPS 2022).

question might be asked: *"Does there exist an active learning method which queries a set of labeled data, such that all the target models can be effectively trained with them?"*

In this paper, we formally define the active learning for multiple target models problem, and reveal the potential improvement of AL under this novel setting. Based on this insight, we further propose an agnostic disagreement-based selection criterion. Specifically, we first define and analyze the label complexity for both active and passive learning under the setting with multiple target models. This label complexity characterizes the number of labeled examples sufficient to train an $\varepsilon$-good classifier with probability at least $1 - \delta$ for each target model. Moreover, we find that the label complexity of single model has a close relation to that of multiple models under the realizable case, e.g., the former provides an upper bound of the label complexity for multiple models, which also implies the potential improvement of AL under this setting. To further explore the agnostic case, we propose an active selection method DIAM (i.e., DIsagreement-based AL for Multi-models) to effectively select the best examples that are beneficial to all target models. It prefers the data located in the joint disagreement regions of different models, which is expected to have higher potential to reduce the soft version space (i.e., the set of hypotheses with lower errors). Experiments are conducted on the OCR benchmarks to validate the necessity of designing active query method under this practical setting and the effectiveness of the proposed approach. The results show that the DIAM method can significantly reduce the number of queries to achieve a higher mean accuracy for multiple models compared to the traditional active and passive learning methods.

The rest of the paper is organized as follows. related work is first reviewed in the following section, then we formally define the AL for multiple target models problem and provide a general result to bridge the label complexity between single and multiple models. Then, we reveal the potential improvement of AL under this novel setting. After that, an agnostic active selection criterion is proposed and analyzed, followed by the empirical studies. And at last, we conclude this work.

## 2 Related work

Active learning has received much attention in recent years due to the greatly increasing demands of labeling data to effectively train more complex models (e.g., deep models) [25]. One of the cores of AL is how to evaluate the potential contribution to the performance improvement of the target model for each candidate query. Most of existing criteria for active learning can be categorized into informativeness and representativeness. The informativeness-based methods [13, 36, 17] prefer the data which is near the decision boundary, and the representativeness-based methods [27, 30, 21] impose the constraints to regularize the queried data to be dissimilar with each other or conform to the latent data distribution. Many works also try to combine both criteria to obtain better performances [11, 37, 31]. Beyond these hand-crafted selection criteria, several meta-active-learning methods [18, 23, 35] are proposed to learn a generalizable query strategy across tasks. Most of the existing active learning query strategies target on improving one specific target model. They are less applicable to the multiple target models setting.

From the theoretical view, active learning theory has also been widely studied under certain conditions (e.g., binary classification, finite VC dimension) [16]. One of the interested properties of an active learning algorithm is the label complexity, which characterizes the number of queries needed to obtain an $\varepsilon$-good classifier with probability at least $1 - \delta$ [14]. To bound this value, disagreement coefficient [5, 4] and Shattering [15, 7] are two commonly used techniques. While most works deal with the single model setting, Balcan *et al.* [3] study the label complexity of the hypothesis space and its subclasses, which sheds light on this work. However, they mainly focus on how to construct subclasses to achieve a certain label complexity, while we aim to find an effective active learning algorithm on the given target models.

Recently, some AL studies tackle a related problem that the target model prior cannot be obtained. In this setting, they will not only search the appropriate target model for the current task, but also avoid noneffective querying. To this end, ALMS [1] either randomly labels data to calculate the unbiased validation error for model selection, or queries by the expected error reduction to improve the models. Active-iNAS [12] considers the deep learning setting, the authors on one hand perform Neural Architecture Search (NAS) to find the appropriate model architecture, on the other hand query the examples by the searched network. Recently, Tang and Huang [32] propose a unified framework DUAL to solve this problem. They query the data that is beneficial to not only the winner model, but

also the model search to identify the high potential model with least queries. All these methods try to search effective model configurations, but not improve multiple target models, which are different from our work.

# 3 Label Complexity of Single Model and Multiple Models

## 3.1 Notations and Definitions

Suppose the data is sampled from an unknown distribution $\mathcal{D}_{XY}$ over the feature space $\mathcal{X}$ and label space $\mathcal{Y}$. Denote by $\mathcal{D}_X$ the marginal data distribution, and $\mathcal{D}_Y$ the marginal label distribution. Given a dataset with $n$ instances, which includes a small labeled set $\mathcal{L} = \{(\boldsymbol{x}_i, y_i)\}_{i=1}^{n_l}$ with $n_l$ instances, and a large unlabeled set $\mathcal{U} = \{\boldsymbol{x}_i\}_{i=n_l+1}^{n_l+n_u}$ with $n_u$ instances, where $n_l \ll n_u$ and $n = n_l + n_u$. At each iteration, the active learning method will select a batch of $b$ examples $\mathcal{Q}$ from $\mathcal{U}$ for querying.

In the single model setting, we are given a hypothesis space $\mathcal{C}$ before querying. While in the multiple target models setting, there are $k$ hypothesis spaces $\mathcal{T} = \{\mathcal{C}_i | i = 1, 2, \ldots, k\}$ with $\tilde{\mathcal{T}} = \bigcup_{i=1}^{k} \mathcal{C}_i$, our goal is to actively query a set of examples to output a well-performed hypothesis $\hat{h}_i$ from each $\mathcal{C}_i, \forall i = 1, ..., k$. We define the true error of a hypothesis as $\mathrm{er}(h) = \mathbb{P}_{\boldsymbol{x} \sim \mathcal{D}_X}(h(\boldsymbol{x}) \neq h^*(\boldsymbol{x}))$, where $h^*$ is the target concept, $h(\boldsymbol{x})$ is the model prediction on the data $\boldsymbol{x}$. The empirical error of $h$ on $\mathcal{L}$ is defined by $\mathrm{er}_{\mathcal{L}}(h) = \frac{1}{|\mathcal{L}|} \sum_{\boldsymbol{x} \in \mathcal{L}} \mathbb{I}[h(\boldsymbol{x}) \neq h^*(\boldsymbol{x})]$, where $\mathbb{I}[\cdot]$ is the indicator function. Let $\nu = \min_{h \in \mathcal{C}} er(h)$, and $\nu_i = \min_{h \in \mathcal{C}_i} er(h)$, $\mathrm{Log}(a) = \max\{\ln(a), 1\}, \forall a > 0$.

Here we introduce the definition of the pseudo-metric between hypotheses, which is frequently used in the subsequent proof.

**Definition 1.** *Pseudo-metric between Hypotheses: Given $\mathcal{D}_X$, the probability of disagreement between two classifiers $h_1$ and $h_2$ is defined as $d(h_1, h_2) = \mathbb{P}_{\boldsymbol{x} \sim \mathcal{D}_X}(h_1(\boldsymbol{x}) \neq h_2(\boldsymbol{x}))$.*

Then we introduce the label complexity of AL for single target model [16].

**Definition 2.** *Label Complexity of AL for Single Target Model: For any active learning algorithm $\mathcal{A}$, we say $\mathcal{A}$ achieves label complexity $\Lambda$ on the hypothesis space $\mathcal{C}$ if, for every $\varepsilon, \delta \in (0, 1)$, every distribution $\mathcal{D}_{XY}$ over $\mathcal{X} \times \mathcal{Y}$, and every integer $t \geq \Lambda(\varepsilon, \delta, \mathcal{D}_{XY})$, if $h_{t,\delta}$ is the classifier produced by running $\mathcal{A}$ with budget $t$, then with probability at least $1 - \delta$, $\mathrm{er}(h_{t,\delta}) - \nu \leq \varepsilon$.*

Now we formally define the label complexity of active learning for multiple target models. It is defined on multiple hypothesis spaces, and the goal is to output an $\varepsilon$-good classifier for each target model. Specifically, the label complexity for the AL with multiple target models is defined as

**Definition 3.** *Label Complexity of AL for Multiple Target Models: Given a set of target models $\mathcal{T} = \{\mathcal{C}_i | i = 1, 2, \ldots, k\}$. For any active learning algorithm $\mathcal{A}$, we say $\mathcal{A}$ achieves label complexity $\tilde{\Lambda}$ for multiple target models if, for every $\varepsilon, \delta \in (0, 1)$, every distribution $\mathcal{D}_{XY}$ over $\mathcal{X} \times \mathcal{Y}$, and every integer $t \geq \tilde{\Lambda}(\varepsilon, \delta, \mathcal{D}_{XY}, \mathcal{T})$, if $\{h_i^{t,\delta} \in \mathcal{C}_i | i = 1, \ldots, k\}$ is the classifiers produced by running $\mathcal{A}$ with budget $t$, then with probability at least $1 - \delta$, $\mathrm{er}(h_i^{t,\delta}) - \nu_i \leq \varepsilon, \forall i = 1, \ldots, k$.*

In the following, we take the passive learning (PL), i.e., random sampling, as a trivial case of active learning, and use the notations $\Lambda^{AL}, \Lambda^{PL}$ and $\tilde{\Lambda}^{AL}, \tilde{\Lambda}^{PL}$ to distinguish the label complexity of them, respectively. We hide the superscript when the context is clear.

## 3.2 Translating the Label Complexity of Single Model to Multiple Models

Denote by $\Lambda_i$ the label complexity of the $i$-th target model $\mathcal{C}_i$. Trivially, the AL label complexity for multiple models has $\tilde{\Lambda}^{AL} \leq \sum_i \Lambda_i^{AL}$ (applying the AL algorithm $\mathcal{A}$ on each of the target model $i$ to get the result). For the passive learning, however, its label complexity for multiple models has a much tighter upper bound $\tilde{\Lambda}^{PL} \leq \max_i \Lambda_i^{PL}$. Because the data is randomly sampled, if $t \geq \max_i \Lambda_i^{PL}(\varepsilon, \delta, \mathcal{D}_{XY})$ examples are queried, according to the definition of label complexity, passive learning will output an $\varepsilon$-good classifier with probability at least $1 - \delta$ for each target model. Such result implicitly indicates that the AL can hardly outperform PL under this setting.

To break this curse, the following theorem is provided to show that, we can expect a much better $\tilde{\Lambda}^{AL}$ for AL under the realizable case (i.e., $h^*$ is in the combined hypothesis space $\tilde{\mathcal{T}}$). It generally says

that, given arbitrary set of target models $\mathcal{T} = \{\mathcal{C}_i | i = 1, 2, \ldots, k\}$, if a learning method has label complexity $\Lambda^{AL}$ on the combined hypothesis space $\tilde{\mathcal{T}}$, it also has the ability to output good classifiers for each $\mathcal{C}_i$, i.e., after querying at most $t$ examples to output $\varepsilon$-good classifiers with probability at least $1 - \delta$ for each $\mathcal{C}_i$.

**Theorem 1.** *Considering binary classification tasks and realizable case, given target models $\mathcal{T} = \{\mathcal{C}_i | i = 1, 2, \ldots, k\}$, assume that active learning algorithm $\mathcal{A}$ achieves label complexity $\Lambda$ on $\tilde{\mathcal{T}}$. Then, there exists an active learning algorithm $\mathcal{A}'$ which achieves the label complexity $\tilde{\Lambda}$ such that $\tilde{\Lambda}(\varepsilon, \delta, \mathcal{D}_{XY}, \mathcal{T}) = \Lambda(\varepsilon/2, \delta, \mathcal{D}_{XY})$.*

*Proof.* Define an algorithm $\mathcal{A}'$ that can output the required classifier $\hat{h}_i \in \mathcal{C}_i, \forall i = 1, \ldots, k$ as follows. First, run the algorithm $\mathcal{A}$ on $(\tilde{\mathcal{T}}, \mathcal{D}_{XY})$ to query $t \geq \Lambda(\varepsilon/2, \delta, \mathcal{D}_{XY})$ labels and output a classifier $h_A$. According to the definition, $d(h_A, h^*)$ is bounded by $\varepsilon/2$ with probability at least $1 - \delta$. Then, for any $\mathcal{C}_i$, output the classifier $\hat{h}_i \in \mathcal{C}_i$ such that $\hat{h}_i = \arg\min_{h_i \in \mathcal{C}_i} d(h_i, h_A)$. Next, we prove that $\mathrm{er}(\hat{h}_i) - \nu_i \leq \varepsilon$ holds with probability at least $1 - \delta$.

To bound $\mathrm{er}(\hat{h}_i)$, it is equivalent to bound $d(\hat{h}_i, h^*)$ by Definition 1. Let $h_i^* = \arg\min_{h_i \in \mathcal{C}_i} \mathrm{er}(h_i)$. It is easy to verify that, $d(\cdot)$ satisfies triangle inequality in binary classification problems, i.e.,

$$d(\hat{h}_i, h^*) \leq d(\hat{h}_i, h_A) + d(h_A, h^*). \tag{1}$$

For the $d(\hat{h}_i, h_A)$, we know that $\hat{h}_i = \arg\min_{h_i \in \mathcal{C}_i} d(h_i, h_A)$, which means

$$d(\hat{h}_i, h_A) \leq d(h_i^*, h_A). \tag{2}$$

Again, by the triangle inequality, we have

$$d(h_i^*, h_A) \leq d(h_i^*, h^*) + d(h^*, h_A). \tag{3}$$

Combining Eq. (1)(2)(3), we have

$$d(\hat{h}_i, h^*) \leq d(h_i^*, h^*) + 2d(h_A, h^*). \tag{4}$$

Since $d(h_A, h^*)$ is bounded by $\varepsilon/2$ with probability at least $1 - \delta$, we can get $\mathrm{er}(\hat{h}_i) - \nu_i \leq \varepsilon$ holds with probability at least $1 - \delta$. $\qquad\square$

Theorem 1 says that if we can find an active learning method to obtain a classifier $h_A \in \tilde{\mathcal{T}}$ such that $\mathrm{er}(h_A) \leq \varepsilon/2$ with probability $1 - \delta$, then we can obtain $\varepsilon$-good classifier $\hat{h}_i$ with probability at least $1 - \delta$ for $\mathcal{C}_i, \forall i = 1, \ldots, k$, where $\hat{h}_i = \arg\min_{h_i \in \mathcal{C}_i} d(h_i, h_A)$. This result provides a general guarantee that if an algorithm can achieve a label complexity on the combined hypothesis space of different models, it also can achieve a bounded label complexity on these models (i.e., the label complexity for multiple models). It can be served as a baseline of AL under multi-models setting.

## 4 Potential Improvements of Active over Passive

Although Theorem 1 bridges the traditional label complexity to that of multiple models setting, it does not reveal the improvement of active over passive learning. Next, we will show the potential of AL under this setting in the realizable case.

According to the theoretical analysis of the passive learning algorithm empirical risk minimization (ERM) [16] for single hypothesis space $\mathcal{C}$ with VC dimension $d$ [34], we know that

**Lemma 1.** *Considering the binary classification, given the hypothesis space $\mathcal{C}$ with VC dimension $d$. The passive learning algorithm $\mathrm{ERM}$ achieves a label complexity $\Lambda^{PL}$ such that, for any $\mathcal{D}_{XY}$ in the realizable case, $\forall \varepsilon, \delta \in (0, 1)$,*

$$\Lambda^{PL}(\varepsilon, \delta, \mathcal{D}_{XY}) \lesssim \left(\frac{1}{\varepsilon}\right)(d \operatorname{Log}(\theta(\varepsilon)) + \operatorname{Log}(1/\delta)). \tag{5}$$

*For the agnostic case, $\mathrm{ERM}$ achieves a label complexity $\Lambda^{PL}$ such that,*

$$\Lambda^{PL}(\nu + \varepsilon, \delta, \mathcal{D}_{XY}) \lesssim \left(\frac{\nu + \varepsilon}{\varepsilon^2}\right)(d \operatorname{Log}(\theta(\nu + \varepsilon)) + \operatorname{Log}(1/\delta)), \tag{6}$$

where $\theta(\cdot)$ is the disagreement coefficient which is formally defined as

**Definition 4.** *Disagreement Coefficient: For any $r_0 \geq 0$ and classifier $h$, define the disagreement coefficient of $h$ with respect to $\mathcal{C}$ on $\mathcal{D}_{XY}$ as*

$$\theta_h^{\mathcal{C}}(r_0) = \sup_{r > r_0} \frac{\mathbb{P}(\mathrm{DIS}(\mathrm{B}_{\mathcal{C}}(h, r)))}{r} \vee 1.$$

*Where $\vee$ is the max operator. For a set of hypotheses $\mathcal{H}$, $\mathrm{DIS}(\mathcal{H}) = \{ \boldsymbol{x} \in \mathcal{X} \mid \exists h, h' \in \mathcal{H}, \ s.t. \ h(\boldsymbol{x}) \neq h'(\boldsymbol{x}) \}$, and $\mathrm{B}_{\mathcal{H}}(h, r) = \{ g \in \mathcal{H} \mid d(h, g) \leq r \}$.*

This value roughly characterizes the behavior of the size of disagreement region $\mathrm{DIS}(\cdot)$ as a function of the hypotheses within a radius $r$ around the classifier $h$. As aforementioned, passive learning has the label complexity for multiple models $\tilde{\Lambda}^{PL} \leq \max_i \Lambda_i^{PL}$. We note that the target concept $h^*$ will usually not be included by every hypothesis space $\mathcal{C}_i$, thus its label complexity $\tilde{\Lambda}^{PL}$ will usually be the agnostic form in Lemma 1 under the realizable case.

To show the potential of AL under this setting, we take the CAL method [9] as an example, which is a representative and well-analyzed approach in the active learning literature [16]. CAL queries the examples from the disagreement region of a set of consistent hypotheses, i.e., $\mathrm{DIS}(V) = \{ \boldsymbol{x} \in \mathcal{X} \mid \exists h, h' \in V \ s.t. \ h(\boldsymbol{x}) \neq h'(\boldsymbol{x}) \}$, where $V = \{ h \in \mathcal{C} \mid h(\boldsymbol{x}) = h^*(\boldsymbol{x}), \forall \boldsymbol{x} \in \mathcal{L} \}$. It achieves the label complexity $O(\theta(\varepsilon) \log(1/\varepsilon) \log(\theta(\varepsilon) \log(1/\varepsilon)))$ for the realizable case. According to Theorem 1, it will have the following label complexity for the multiple target models

**Corollary 1.** *Given target models $\mathcal{T} = \{ \mathcal{C}_i \mid i = 1, 2, \ldots, k \}$. Suppose $\tilde{\mathcal{T}}$ has VC dimension $d < \infty$. CAL achieves a label complexity $\tilde{\Lambda}^{AL}$ for multiple target models such that, for $\mathcal{D}_{XY}$ in the realizable case, for any $\forall \varepsilon, \delta \in (0, 1)$,*

$$\tilde{\Lambda}^{AL}(\varepsilon, \delta, \mathcal{D}_{XY}, \mathcal{T}) \leq \theta_{h^*}^{\tilde{\mathcal{T}}}(\varepsilon/2) \mathrm{Log}(2/\varepsilon) \left( d \, \mathrm{Log}(\theta_{h^*}^{\tilde{\mathcal{T}}}(\varepsilon/2)) + \mathrm{Log}\left( \frac{\mathrm{Log}(2/\varepsilon)}{\delta} \right) \right). \tag{7}$$

*Proof.* By combining the label complexity of CAL for single model from [16] and Theorem 1, we can get the result. $\qquad\square$

To reveal the potential improvement, note that the label complexity for passive learning heavily depends on the property of the worst hypothesis space, i.e., the value of $\max_i \min_{h \in \mathcal{C}_i} er(h)$. Assume that $\max_i \min_{h \in \mathcal{C}_i} er(h) > \varepsilon$. Then according to Lemma 1 and Corollary 1, the label complexity of passive learning for multiple target models $\tilde{\Lambda}^{PL}(\varepsilon, \delta, \mathcal{D}_{XY}, \mathcal{T})$ is $\Omega(2/\varepsilon)$. On the other side, CAL has a label complexity $\Omega(\mathrm{Log}(2/\varepsilon))$, which implies the potential of the improvement of active learning under this setting. We leave the guarantee of strict improvement of AL under this setting an interesting future work. Next we further study the agnostic case (i.e., $h^* \notin \tilde{\mathcal{T}}$).

## 5 An Agnostic Disagreement-based AL method for Multiple Models

Define the set $V_i$ for each $\mathcal{C}_i$ as $\{ h \in \mathcal{C}_i \mid h(\boldsymbol{x}) = h^*(\boldsymbol{x}), \forall \boldsymbol{x} \in \mathcal{L} \}$, we propose to query the examples located in the joint disagreement regions for all $\mathcal{C}_i, \forall i = 1, 2, \ldots, k$, i.e., $\mathrm{DIS}(V_1) \cap \mathrm{DIS}(V_2) \cap \ldots \mathrm{DIS}(V_k)$. Intuitively, we know that $V_i$ must be a subset of $V$, if such data exists, we can expect it has higher potential to reduce $V$. This statement can be simply implied by the Bayesian formula.

**Proposition 1.** *Considering binary classification problem. Given hypothesis space $\mathcal{C}$. Let $V_+(\boldsymbol{x}) = \{ h \in V \mid h(\boldsymbol{x}) = +1 \}$, $V_-(\boldsymbol{x}) = \{ h \in V \mid h(\boldsymbol{x}) = -1 \}$, where $V = \{ h \in \mathcal{C} \mid h(\boldsymbol{x}) = h^*(\boldsymbol{x}), \forall \boldsymbol{x} \in \mathcal{L} \}$. Denote $\lambda(\boldsymbol{x}) = \frac{|V_+(\boldsymbol{x})|}{|V_-(\boldsymbol{x})|}$, where $|\cdot|$ is the number of elements of a set. The ideal case is to query the $\boldsymbol{x}$ which has $\lambda(\boldsymbol{x}) = 1$. Given any sequences of subset $V_1, V_2, \ldots, V_k$ randomly sampled from $V$, define the event $E_{\boldsymbol{x}}$ that data $\boldsymbol{x}$ falls into $\mathrm{DIS}(V_1) \cap \mathrm{DIS}(V_2) \cdots \cap \mathrm{DIS}(V_k)$. According to the Bayesian formula, we have*

$$\mathbb{P}(\lambda(\boldsymbol{x}) = 1 | E_{\boldsymbol{x}}) = \frac{\mathbb{P}(E_{\boldsymbol{x}} | \lambda(\boldsymbol{x}) = 1) \mathbb{P}(\lambda(\boldsymbol{x}) = 1)}{\mathbb{P}(E_{\boldsymbol{x}})}$$
$$\geq \mathbb{P}(\lambda(\boldsymbol{x}) = 1). \tag{8}$$

| **Algorithm 1** The DIAM-online Algorithm | **Algorithm 2** The DIAM-pool Algorithm |
|---|---|
| **Initialize:** hyperparameter $q$, constants $\sigma_i$; $m \leftarrow 0, \hat{V}_i \leftarrow \mathcal{C}_i, \forall i = 1, \ldots, k$. 
 **Output:** Any $h \in \hat{V}_i, \forall i = 1, \ldots, k$. | **Initialize:** labeled set $\mathcal{L}$, unlabeled set $\mathcal{U}$, hyperparameters $\hat{\sigma}_i, \hat{V}_i \leftarrow \mathcal{C}_i, \forall i = 1, \ldots, k$. 
 **Output:** $\{\hat{h}_i | i = 1, \ldots, k\}$. |

| | |
|---|---|
| 1: **while** Labeling budget is not run out **do** | 1: **while** Labeling budget is not run out **do** |
| 2: $\quad m \leftarrow m + 1$ | 2: $\quad \boldsymbol{x}^* = \arg\max_{\boldsymbol{x} \in \mathcal{U}} \sum_i \mathbb{I}[\boldsymbol{x} \in \mathrm{DIS}(\hat{V}_i)]$ |
| 3: $\quad$ Request an unlabeled data $\boldsymbol{x}_m$ | 3: $\quad$ Query $\boldsymbol{x}^*$ from the oralce: $\mathcal{L} \leftarrow \mathcal{L} \cup$ |
| 4: $\quad$ **if** $\sum_i \mathbb{I}[\boldsymbol{x}_m \in \mathrm{DIS}(\hat{V}_i)] \geq q$ **then** | $\quad \{(\boldsymbol{x}^*, h^*(\boldsymbol{x}^*)))\}$ |
| 5: $\quad\quad$ Query: $\mathcal{L} \leftarrow \mathcal{L} \cup \{(\boldsymbol{x}_m, h^*(\boldsymbol{x}_m))\}$ | 4: $\quad \mathcal{U} \leftarrow \mathcal{U} \setminus \{\boldsymbol{x}^*\}$ |
| 6: $\quad$ **end if** | 5: $\quad$ **for** $i = 1, \ldots, k$ **do** |
| 7: $\quad$ **if** $\log_2 m \in \mathbb{N}$ **then** | 6: $\quad\quad \hat{h}_i \leftarrow \min_{g \in \hat{V}_i} \mathrm{er}_{\mathcal{L}}(g)$ |
| 8: $\quad\quad \hat{V}_i \leftarrow \{h \in \hat{V}_i | \mathrm{er}_{\mathcal{L}}(h) - \min_{g \in \hat{V}_i} \mathrm{er}_{\mathcal{L}}(g) \leq \sigma_i\}, \forall i = 1, \ldots, k.$ | 7: $\quad\quad \hat{V}_i \leftarrow \{h \in \hat{V}_i | (\mathrm{er}_{\mathcal{L}}(h) - \hat{h}_i) \leq \hat{\sigma}_i\}$ |
| 9: $\quad$ **end if** | 8: $\quad$ **end for** |
| 10: **end while** | 9: **end while** |

*Proof.* Since each $V_i$ is randomly sampled from $V$, $\mathbb{P}(E_{\boldsymbol{x}})$ will reach its maximum value when $\lambda(\boldsymbol{x}) = 1$, thus we have $\mathbb{P}(E_{\boldsymbol{x}}|\lambda(\boldsymbol{x}) = 1)/\mathbb{P}(E_{\boldsymbol{x}}) \geq 1$, which leads to the conclusion. $\qquad\square$

Following this principle, we would like to query the examples located in the joint disagreement regions of $\mathcal{C}_i, \forall i = 1, 2, \ldots, k$. However, since we have multiple target models, the target concept $h^*$ might not be included by every $\mathcal{C}_i$ in practice, which turns the learning problem to the agnostic setting. Inspired by the RobustCAL method [2], which is a disagreement-based AL algorithm for the agnostic setting, we propose DIAM (i.e., DIsagreement-based AL for Multi-models) query strategy for the multiple target models problem. Note that we define a new form of $V_i$ as $\hat{V}_i$ to tackle the noisy setting, i.e., $\hat{V}_i = \{h \in \mathcal{C}_i \mid \mathrm{er}_{\mathcal{L}}(h) - \min_{g \in \mathcal{C}_i} \mathrm{er}_{\mathcal{L}}(g) \leq \sigma_i\}$, where $\sigma_i$ is a constant. To simplify the theoretical analysis, we first propose an online version of DIAM, then we define the DIAM method for the pool-based setting and empirically validate its effectiveness. They are summarized at Algorithm 1 and 2, respectively. The hyperparameter $q$ controls the conservativeness of the algorithm. With a larger $q$, it will reject more less informative unlabeled data in the online setting.

Now let us analyze the DIAM method. Since we are considering the agnostic setting, it is necessary to model the noise. Here we employ the commonly used Tsybakov noise condition [33].

**Condition 1.** *[33, Tsybakov noise] For some $a \in [1, \infty)$ and $\alpha \in [0, 1]$, assume that $f^\star$ achieves $\inf_{h \in \mathcal{C}} \mathrm{er}(h)$, for every $h \in \mathcal{C}$,*

$$\mathbb{P}\left(\boldsymbol{x} : h(\boldsymbol{x}) \neq f^\star(\boldsymbol{x})\right) \leq a \left(\mathrm{er}(h) - \mathrm{er}\left(f^\star\right)\right)^\alpha.$$

We assume that there exists pair of $a_i$ and $\alpha_i$ for each target model $\mathcal{C}_i$. Considering a conservative situation that the hyperparameter $q = 1$, by further taking the constants $\sigma_i$ in DIAM-online algorithm as the same form in the RobustCAL method [16], which relates to the properties of the noise, hypothesis space, and disagreement coefficient, we can have the following result. The proof is deferred to the appendix.

**Theorem 2.** *Considering agnostic setting and binary classification tasks. Given a set of target models $\mathcal{T} = \{\mathcal{C}_i | i = 1, 2, \ldots, k\}$, in which each $\mathcal{C}_i$ has VC dimensions $d_i < \infty$ and meet Condition 1. Let $h_i^* = \arg\min_{h_i \in \mathcal{C}_i} \mathrm{er}(h_i)$. For any $\varepsilon, \delta \in (0, 1)$, if $q = 1$, DIAM-online algorithm achieves a label complexity $\tilde{\Lambda}(\varepsilon, \delta, \mathcal{D}_{XY}, \mathcal{T})$ for multiple target models such that, for $a_i$ and $\alpha_i$ as in Condition 1, for any $\mathcal{D}_{XY}, \Lambda(\varepsilon, \delta, \mathcal{D}_{XY}, \mathcal{T})$ is no larger than*

$$\sum_{i=1}^k a_i^2 \theta_{h_i^*}^{\mathcal{C}_i}\left(a_i \varepsilon^{\alpha_i}\right) \varepsilon^{2\alpha_i - 2} \left(d_i \mathrm{Log}\left(\theta_{h_i^*}^{\mathcal{C}_i}\left(a_i \varepsilon^{\alpha_i}\right)\right) + \mathrm{Log}\left(\frac{\mathrm{Log}(a_i/\varepsilon)}{\delta}\right)\right) \mathrm{Log}(\frac{1}{\varepsilon}), \qquad (9)$$

*and no larger than,*

$$\sum_{i=1}^k \theta_{h_i^*}^{\mathcal{C}_i}(\nu_i + \varepsilon) \left(\frac{\nu_i^2}{\varepsilon^2} + \mathrm{Log}\left(\frac{1}{\varepsilon}\right)\right) \left(d_i \mathrm{Log}(\theta_{h_i^*}^{\mathcal{C}_i}(\nu_i + \varepsilon)) + \mathrm{Log}\left(\frac{\mathrm{Log}(1/\varepsilon)}{\delta}\right)\right). \qquad (10)$$

Theorem 2 provides an upper bound of the label complexity of the DIAM-online method when $q = 1$. It considers a general situation with arbitrary target models and data distributions, even the unlabeled data will never fall into the joint disagreement regions. However, one may be more interested in the situation that if we can always query the $x$ such that $\sum_i \mathbb{I}[x \in \text{DIS}(\hat{V}_i)] = k$. Next, we prove that if such ideal situation exists, DIAM-online will achieve a better label complexity than applying CAL on the multiple target models setting even under the realizable setting.

**Theorem 3.** *Considering binary classification tasks and realizable case. Given a set of target models $\mathcal{T} = \{\mathcal{C}_i | i = 1, 2, \ldots, k\}$, in which each $\mathcal{C}_i$ has VC dimensions $d_i < \infty$ and meet Condition 1, and $\tilde{\mathcal{T}}$ with VC dimension $d < \infty$. Assume that, if a data point falls into the disagreement region of any $\mathcal{C}_i$, it also falls into the disagreement regions of the others $\{\mathcal{C}_j | j \neq i, j = 1, 2, \ldots, k\}$. Assume the $m$-th target model achieves the highest label complexity. Let $h_m^* = \arg\min_{h_m \in \mathcal{C}_m} \text{er}(h_m)$ and $\nu_m = \text{er}(h_m^*)$. For any $\delta \in (0, 1)$, $\varepsilon \in (0, 1/e)$, $h^* \in \tilde{\mathcal{T}}$. If $\nu_m \leq \frac{\ln 2}{2}\varepsilon$, DIAM-online achieves a better upper bound of $\tilde{\Lambda}$ than that of applying CAL method on $\tilde{\mathcal{T}}$.*

The key of the proof is comparing the disagreement coefficients defined on different functions and hypothesis spaces, i.e., $\theta_{h_m^*}^{\mathcal{C}_m}$ and $\theta_{h^*}^{\tilde{\mathcal{T}}}$. We defer the proof to the appendix. Although Theorem 3 holds with somewhat strict conditions, we note that Theorem 1 only works in the realizable case, while DIAM does not require this condition. Next, we discuss how to implement DIAM in the real applications for deep models.

It is generally believed that finding the disagreed pair of classifiers from a set of hypotheses for a given $x$ is non-trivial. Most existing methods randomly sample functions from the hypothesis space for validation, or turn to select the data close to the decision boundary (e.g., uncertainty), which can be expensive or inaccurate. This problem becomes more prohibitive to the deep models.

To efficiently estimate the disagreement regions for the neural networks, we propose to exploit the predictions of the unlabeled data during the later epochs in the training phase, typically after the network converging. Recall the definition of disagreement region $\text{DIS}(\hat{V}_i)$, we should firstly find the hypotheses that are basically consistent with the labeled data, then validate whether there exists a pair of hypotheses that disagree on the given unlabeled data. For the first characteristic, the models on the later epochs, i.e., has smaller training errors, can represent the well-learned hypotheses. For the second characteristic, if there exists models $i, j$ from the later epochs such that the model trained at epoch $i$ has inconsistent prediction with the model trained at epoch $j$ on the unlabeled data $x$, we can say that the example $x$ falls into $\text{DIS}(\hat{V}_i)$.

More concretely, according to the training loss curve, we empirically take the models in the latter half of the training epochs as the well-performed hypotheses set, and estimate the disagreement region with them. Since there may be multiple examples have the same value of $\sum_i \mathbb{I}[x \in \text{DIS}(\hat{V}_i)]$, we further calculate the vote entropy [10] of the well-performed hypotheses on the specific data, and take it as the secondary sort key in the data selection phase. We also note that the query batch size in deep learning is usually large. tT avoid overmuch information redundancy, we heuristically keep the top-rated unlabeled data with 5 times the size of the batch size, and randomly sample $20\%$ from it for querying. We leave the more advanced diversity measurement a future work.

By applying the above heuristics to the deep models, DIAM method is quite efficient. It evaluates the unlabeled data with the models trained with later epochs, which roughly takes the size of the well-performed hypotheses set times of that of the entropy method to make the data selection. However, we also note that the DIAM method has some limitations. It only considers the informativeness of the unlabeled data, which may be less effective for the batch-mode selection. For the potential negative social impact, DIAM may reduce the cost of training multiple malicious machine learning models. Nevertheless, we believe the positive contribution is more significant.

## 6 Experiment

### 6.1 Empirical Settings

To construct the multiple target models scenario, we introduce the results of a recent NAS method OFA [6], which tries to efficiently search model architectures for different devices by training only one super-net. They report the searched effective architectures that meet the hardware constraints

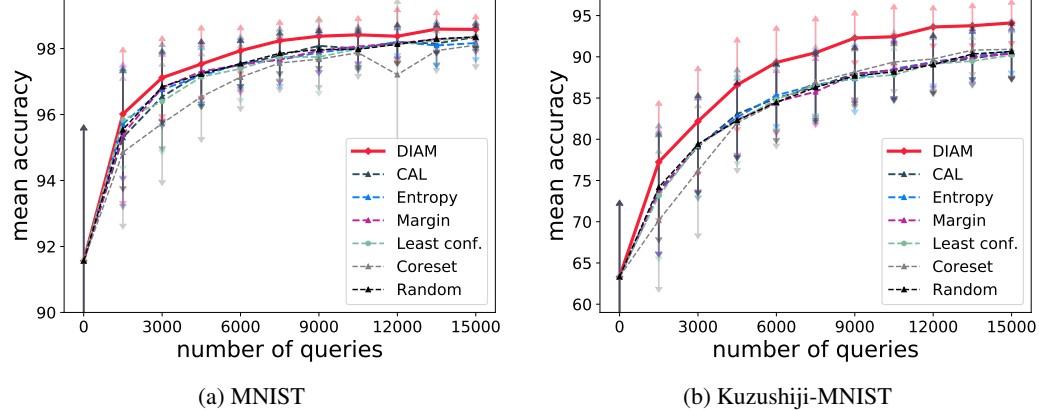

|       |       |
|-------|-------|
| (a) MNIST | (b) Kuzushiji-MNIST |

Figure 1: The learning curves with the mean accuracy of the target models of the compared methods. The error bars indicate the standard deviation of the performances of target models.

of various machines on the GitHub [2], which is well suited to our problem setting. Specifically, we take 12 specialized model architectures with different prediction accuracies and speeds that target on Samsung S7 Edge, Samsung Note8 and Samsung Note10 as our target models. They are pruned from a MobileNetV3 (which is the super-net), but have very different prediction time and accuracies. Their Multiply-Accumulate Operations (MACs) range from 66M to 237M, which denote the diversity of the architectures. The model specifications are listed in the appendix.

We compare the following query strategies in our experiments.

- DIAM: The proposed method of this paper, which queries the data located in the joint disagreement regions of multiple target models.

- CAL [9]: Query the data falls into the disagreement region of any target models. It has a bounded label complexity for the multiple target models setting according to Theorem 1.

- Entropy [20]: Query the data with the highest prediction entropies. We take the mean entropies calculated by all target models to support the novel problem setting.

- Least Confidence [29]: Query the data with the least prediction confidence. We take the mean values calculated by all target models to support the novel problem setting.

- Margin [26]: Query the data with the minimum prediction margin. We take the mean margin values calculated by all target models to support the novel problem setting.

- Coreset [27]: Query the most representative data. The distance is calculated by the features extracted by a pretrained MobileNetV3, which is the super-net in OFA [6].

- Random: Query data randomly. Note that this is a highly competitive baseline.

Since Optical Character Recognition (OCR) is one of the representative machine learning systems that are required to be deployed on diverse devices, two commonly used hand-writing characters classification benchmarks are employed in our experiments, i.e., the MNIST [19] and Kuzushiji-MNIST [8] datasets. They are under the CC BY-SA 3.0 and CC BY-SA 4.0 licenses, respectively. Here we consider the prevalent pool-based active learning setting. Specifically, we randomly take $3,000$ training data as our initially labeled data, and the rest as the unlabeled pool. At each iteration, the compared sampling methods will select $1,500$ unlabeled examples for querying, then re-train the models. The mean and standard deviation of the accuracies of multiple target models are reported. Note that more results can be found in the appendix.

For the model training, We mainly follow the training configs of OFA. Specifically, the hyperparameters are set by the default values in the project. For example, the learning rate is set by $7.5e-3$, batch size is $128$, SGD optimizer is employed with momentum $0.9$. Since the initially labeled data is limited, a small number of training epochs is taken to avoid over-fitting. Specifically, we employ the

---

[2] https://github.com/mit-han-lab/once-for-all . It is under the MIT license.

Table 1: The mean of the learning curves and the mean of standard deviation values with different numbers of target models on the OCR benchmarks achieved by the compared methods (mean accuracy ± mean standard deviation). The best performance is highlighted in boldface.

| Methods | Number of Target Models | | | | |
|---|---|---|---|---|---|
| | 2 | 4 | 6 | 8 | 12 |
| MNIST | | | | | |
| DIAM | **98.16 ± 0.13** | **97.29 ± 0.99** | **97.55 ± 0.85** | **97.34 ± 1.09** | **97.34 ± 1.04** |
| CAL | 97.79 ± 0.14 | 97.04 ± 0.92 | 97.24 ± 0.89 | 96.95 ± 1.07 | 96.98 ± 1.10 |
| Entropy | 97.83 ± 0.10 | 96.94 ± 1.01 | 97.15 ± 0.98 | 96.92 ± 1.06 | 96.98 ± 1.00 |
| Margin | 97.79 ± 0.13 | 96.94 ± 1.02 | 97.19 ± 0.96 | 96.81 ± 1.19 | 97.00 ± 1.02 |
| Least conf. | 97.84 ± 0.11 | 96.89 ± 1.02 | 97.23 ± 0.92 | 96.88 ± 1.05 | 96.96 ± 1.07 |
| Coreset | 97.64 ± 0.13 | 96.69 ± 1.07 | 97.03 ± 0.97 | 96.36 ± 1.82 | 96.56 ± 1.40 |
| Random | 97.81 ± 0.12 | 96.93 ± 0.97 | 97.21 ± 0.94 | 96.83 ± 1.12 | 97.03 ± 0.99 |
| Kuzushiji-MNIST | | | | | |
| DIAM | **90.38 ± 0.21** | **85.76 ± 4.69** | **86.91 ± 4.38** | **86.23 ± 4.68** | **86.85 ± 4.25** |
| CAL | 87.06 ± 0.34 | 83.61 ± 4.29 | 84.70 ± 3.88 | 83.40 ± 4.32 | 83.31 ± 4.53 |
| Entropy | 87.09 ± 0.34 | 83.22 ± 4.16 | 84.39 ± 3.85 | 83.28 ± 4.28 | 83.33 ± 4.33 |
| Margin | 86.91 ± 0.35 | 83.20 ± 4.10 | 84.31 ± 4.03 | 83.11 ± 4.37 | 83.16 ± 4.29 |
| Least conf. | 86.71 ± 0.26 | 83.38 ± 4.25 | 84.36 ± 3.71 | 83.20 ± 4.42 | 83.04 ± 4.33 |
| Coreset | 87.49 ± 0.36 | 82.97 ± 5.16 | 84.80 ± 4.58 | 83.00 ± 4.93 | 82.91 ± 5.03 |
| Random | 87.34 ± 0.31 | 82.97 ± 4.38 | 84.22 ± 3.98 | 83.02 ± 4.19 | 83.26 ± 4.36 |

pretrained weights on the image-net dataset for initialization, then finetune 20 epochs on the labeled data.

## 6.2 Results

We report the trend of mean accuracy of multiple target models with the number of queries increasing in Fig. 1. The error bars indicate the standard deviation of the performances of multiple target models. First of all, the high deviation of the performances of the initial point shows the diversity of the target models, which symbolizes the practicability and difficulty of the experimental settings. It is conceivable that different target models will have various preferences of training data due to the diverse architectures. Under this challenging setting, it can be observed from the figure that our method can significantly outperform the traditional active and passive learning methods. It shows a great potential of improvements over the random sampling, which is a very competitive baseline in this novel setting. This result sufficiently reveals the effectiveness of DIAM and the necessity of designing active query method under this practical setting. The uncertainty-based methods, i.e., entropy, least confidence and margin, achieve comparable performances with random sampling. These results meet our expectation. Because traditional AL methods are usually model-dependent, i.e., the data queried by one model may be less effective for training another model. By taking the mean uncertainty scores of diverse target models, the data selection may tend to be non-informative. The coreset method is less stable than random. We note that coreset is still a model-based selection method in deep learning. Because the features of the data will be optimized along with the training procedures. Thus it may also suffer from the model dependence problem.

## 6.3 Study on Different Numbers of Target Models

We further explore the influence of the number of target models to the data selection methods. Due to the space limitation, we report the mean of the learning curves and the mean of standard deviations in Table 1, and defer the whole learning curves to the appendix. The results show that our method can consistently outperform the other compared methods, which demonstrate its robustness to the number of models. This property also denotes that the DIAM method has the potential to tackle more challenging situations, i.e., improving sufficient numbers of target models simultaneously. It is essential to the machine learning systems which have a wide range of applications. The performances of the other compared methods have similar trends with more target models setting. It again verifies that the traditional AL methods are usually model-dependent, and emphasizes the necessity of designing novel selection approaches under this practical setting.

# 7 Conclusion

In this paper, we propose to study active learning in a novel setting, where the task is to select and label the most useful examples that are beneficial to multiple target models. We firstly analyze the label complexity of active and passive learning to reveal the potential improvement of AL under this novel setting. Based on this insight, we further propose an active selection criterion DIAM that prefers the data located in the joint disagreement regions of different target models. Empirical studies on the OCR benchmarks, which is one of the representative applications that are required to accommodate different devices, show the effectiveness of the proposed method. In the future, we will tackle more complex and important learning tasks (e.g., face recognition, object detection), and design effective query strategies which incorporate both informativeness and representativeness under the multiple target models setting.

## Acknowledgments and Disclosure of Funding

This research was supported by the National Key R&D Program of China (2020AAA0107000), NSFC (62222605, 62076128), and Natural Science Foundation of Jiangsu Province of China (BK20211517, BK2022050029).

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
