# Active Learning for Multiple Target Models
# (Supplementary Materials)

In the appendix, we will first prove of the theorems in the paper, then we introduce more details and results of the experiments, which include more particular empirical settings, running time, computational resources, the significance of the performance comparisons and extra experimental results. The notations in the following contents are consistent with the paper.

## A  Proofs of the Theorems in the Paper

### A.1  Proof of Theorem 2

The proof of Theorem 2 is based on the results of the RobustCAL method [2]. Thus we first introduce the following results of the RobustCAL method for the single model.

**Lemma i.** *[3, Theorem 5.4] Considering binary classification problem. Suppose the hypothesis space $\mathcal{C}$ has VC dimension $d$. For any $\delta \in (0,1)$, RobustCAL achieves a label complexity $\Lambda$ such that, for any $\mathcal{D}_{XY}$, for $a$ and $\alpha$ as in Condition 1, $\forall \varepsilon \in (0,1)$,*

$$\Lambda\left(\nu + \varepsilon, \delta, \mathcal{P}_{XY}\right) \lesssim a^2 \theta\left(a\varepsilon^\alpha\right) \left(\frac{1}{\varepsilon}\right)^{2-2\alpha} \left(d \operatorname{Log}\left(\theta\left(a\varepsilon^\alpha\right)\right) + \operatorname{Log}\left(\frac{\operatorname{Log}(a/\varepsilon)}{\delta}\right)\right) \operatorname{Log}(1/\varepsilon),$$

*and furthermore,*

$$\Lambda\left(\nu + \varepsilon, \delta, \mathcal{P}_{XY}\right) \lesssim \theta(\nu + \varepsilon) \left(\frac{\nu^2}{\varepsilon^2} + \operatorname{Log}\left(\frac{1}{\varepsilon}\right)\right) \left(d \operatorname{Log}(\theta(\nu + \varepsilon)) + \operatorname{Log}\left(\frac{\operatorname{Log}(1/\varepsilon)}{\delta}\right)\right).$$

With the above techniques, we begin to prove Theorem 2.

*Proof.* The hyperparameter $q = 1$ in the DIAM-online method means querying the data falls into any disagreement regions of target models. In other words, the data queried for a specific target model must be queried by the DIAM-online algorithm. Denote the number of data queried for $i$-th target model by $t_i$ such that it is sufficient to output an $\varepsilon$-good classifier with probability at least $1 - \delta$ for $\mathcal{C}_i$. By a union bound, we know that DIAM-online queries $t \geq \sum_i t_i$ examples sufficient to output $\varepsilon$-good classifiers for each target model with probability at least $1 - \delta$. Recall that DIAM-online method takes the same form of $\sigma_i$ with the RobustCAL method (step 8 in the Algorithm 1 in the paper), thus it is equivalent to applying RobustCAL on each target model. By incorporating Lemma i and a union bound, we can get the conclusion. $\square$

### A.2  Proof of Theorem 3

To compare the upper bound of the label complexity between DIAM and CAL under the multiple target models setting, we first note that, if the ideal situation described in the Theorem 3 exists, then DIAM-online will achieve the label complexity $\tilde{\Lambda} = \max_i \Lambda_i$. Because the queried data is useful for all hypothesis spaces $\mathcal{C}_i, \forall i = 1, \ldots, k$. Thus, if $t > \max_i \Lambda_i$ examples are labeled, DIAM-online will output the desired classifiers for each $\mathcal{C}_i$. Assume the $m$-th target model achieves the highest label complexity, by Lemma i, we know that DIAM achieves the label complexity for multiple models

36th Conference on Neural Information Processing Systems (NeurIPS 2022).

with $\varepsilon \in (0, 1/e)$ in binary classification problem, such that

$$\tilde{\Lambda} \leq \theta^{\mathcal{C}_m}_{h^*_m}(\nu_m + \varepsilon) \left( \frac{\nu^2_m}{\varepsilon^2} + \ln\left(\frac{1}{\varepsilon}\right) \right) \left( d_m \ln(\theta^{\mathcal{C}_m}_{h^*_m}(\nu_m + \varepsilon)) + \ln\left(\frac{\ln(1/\varepsilon)}{\delta}\right) \right). \tag{1}$$

For the CAL method, according to the Corollary 1 in the paper, we know that it achieves the label complexity for multiple models in binary classification problems such that

$$\tilde{\Lambda} \leq \theta^{\tilde{\mathcal{T}}}_{h^*}(\varepsilon/2) \ln(2/\varepsilon) \left( d \ln(\theta^{\tilde{\mathcal{T}}}_{h^*}(\varepsilon/2)) + \ln\left(\frac{\ln(2/\varepsilon)}{\delta}\right) \right). \tag{2}$$

To compare the right side of Eq. (1) and (2), one challenge is to compare the disagreement coefficients defined on different functions and hypothesis spaces, i.e., $\theta^{\mathcal{C}_m}_{h^*_m}$ and $\theta^{\tilde{\mathcal{T}}}_{h^*}$. To this end, we first introduce the following properties of the disagreement coefficient $\theta(\cdot)$, which are analyzed in [3].

**Lemma ii.** *[3, Theorem 7.1] Given $h \in \mathcal{C}$, $\theta_h(r)$ is nonincreasing w.r.t. $r \in [0, +\infty)$.*

**Lemma iii.** *[3, Theorem 7.8] Let $\mathcal{C}_1$ and $\mathcal{C}_2$ be sets of classifiers such that $\mathcal{C} = \mathcal{C}_1 \cup \mathcal{C}_2$. For all $\varepsilon > 0$, let $\theta^{\mathcal{C}}_h(\varepsilon), \theta^{\mathcal{C}_1}_h(\varepsilon)$, and $\theta^{\mathcal{C}_2}_h(\varepsilon)$ denote the disagreement coefficients of arbitrary $h$ (not necessarily in $\mathcal{C}$) with respect to $\mathcal{C}, \mathcal{C}_1, \mathcal{C}_2$, respectively, under $\mathcal{D}_X$. Then $\forall \varepsilon > 0$,*

$$\max\left\{\theta^{\mathcal{C}_1}_h(\varepsilon), \theta^{\mathcal{C}_2}_h(\varepsilon)\right\} \leq \theta^{\mathcal{C}}_h(\varepsilon).$$

**Lemma iv.** *[3, Corollary 7.2] Let $\varepsilon \in (0, \infty)$ and $a \in (1, \infty)$. Then $\theta^{\mathcal{C}}_h(\varepsilon/a) \leq a\theta^{\mathcal{C}}_h(\varepsilon)$ and $\theta^{\mathcal{C}}_h(\varepsilon)/a \leq \theta^{\mathcal{C}}_h(a\varepsilon)$.*

**Lemma v.** *$\forall h \in \mathcal{C}$, given a hypothesis $g$ (not necessarily in $\mathcal{C}$), if $d(h, g) \leq \gamma$, for any $\gamma > 0$. Then $\forall \varepsilon > 0$ we have*

$$\theta^{\mathcal{C}}_g(\varepsilon) \leq \frac{\varepsilon + \gamma}{\varepsilon} \theta^{\mathcal{C}}_h(\varepsilon + \gamma) \leq \frac{\varepsilon + \gamma}{\varepsilon} \theta^{\mathcal{C}}_h(\varepsilon) \tag{3}$$

*Proof.* (Lemma v) $d(h, g) \leq \gamma$ implies that $\forall r > 0$, $\mathrm{B}_{\mathcal{C}}(g, r + \gamma) \supseteq \mathrm{B}_{\mathcal{C}}(h, r)$ and $\mathrm{B}_{\mathcal{C}}(h, r + \gamma) \supseteq \mathrm{B}_{\mathcal{C}}(g, r)$. Then

$$
\begin{aligned}
\theta^{\mathcal{C}}_g(\varepsilon) &= 1 \vee \sup_{r > \varepsilon} \frac{\mathbb{P}\left(\mathrm{DIS}\left(\mathrm{B}_{\mathcal{C}}\left(g, r\right)\right)\right)}{r} \leq 1 \vee \sup_{r > \varepsilon} \frac{\mathbb{P}(\mathrm{DIS}(\mathrm{B}_{\mathcal{C}}(h, r + \gamma)))}{r} \\
&\leq \frac{\varepsilon + \gamma}{\varepsilon} \left( 1 \vee \sup_{r > \varepsilon} \frac{\mathbb{P}(\mathrm{DIS}(\mathrm{B}_{\mathcal{C}}(h, r + \gamma)))}{r + \gamma} \right) \\
&= \frac{\varepsilon + \gamma}{\varepsilon} \theta^{\mathcal{C}}_h(\varepsilon + \gamma) \leq \frac{\varepsilon + \gamma}{\varepsilon} \theta^{\mathcal{C}}_h(\varepsilon).
\end{aligned} \tag{4}
$$

$\square$

Lemma iii and Lemma v bridge the disagreement coefficients defined on different classifiers and hypothesis spaces. With the above results, we can now begin to prove Theorem 3.

*Proof.* (Theorem 3)

To compare the right side of Eq. (1) and (2), we turn to compare each corresponding term, i.e., $\theta^{\mathcal{C}_m}_{h^*_m}(\nu_m + \varepsilon)$ and $\theta^{\tilde{\mathcal{T}}}_{h^*}(\varepsilon/2)$; $\frac{\nu^2_m}{\varepsilon^2} + \ln\left(\frac{1}{\varepsilon}\right)$ and $\ln\left(\frac{2}{\varepsilon}\right)$.

For the first group of terms (i.e., disagreement coefficients), by Lemma v, we have

$$\theta^{\mathcal{C}_m}_{h^*_m}(\nu_m + \varepsilon) \leq \frac{\varepsilon + 2\nu_m}{\varepsilon + \nu_m} \theta^{\mathcal{C}_m}_{h^*}(\nu_m + \varepsilon). \tag{5}$$

Since $\nu_m \leq \frac{\ln 2}{2}\varepsilon < \frac{1}{2}\varepsilon$, we can get

$$1 < \frac{\varepsilon + 2\nu_m}{\varepsilon + \nu_m} < \frac{4}{3}. \tag{6}$$

According to Lemma iv, we can get

$$\frac{\varepsilon + 2\nu_m}{\varepsilon + \nu_m} \theta^{\mathcal{C}_m}_{h^*}(\nu_m + \varepsilon) \leq \theta^{\mathcal{C}_m}_{h^*}\left(\frac{3}{2}\nu_m + \frac{3}{4}\varepsilon\right). \tag{7}$$

Combining Lemma ii and Lemma iii, we have

$$\theta_{h^*}^{\mathcal{C}_m}(\frac{3}{2}\nu_m + \frac{3}{4}\varepsilon) \le \theta_{h^*}^{\mathcal{C}_m}(\varepsilon/2) \le \theta_{h^*}^{\tilde{\mathcal{T}}}(\varepsilon/2) \,. \tag{8}$$

For the second group of terms (i.e., $\frac{\nu_m^2}{\varepsilon^2} + \ln\left(\frac{1}{\varepsilon}\right)$ and $\ln\left(\frac{2}{\varepsilon}\right)$), according to the assumptions $\nu_m \le \frac{\ln 2}{2}\varepsilon$, we have

$$\frac{\nu_m^2}{\varepsilon^2} + \ln\left(\frac{1}{\varepsilon}\right) \le \ln\left(\frac{2}{\varepsilon}\right) \,. \tag{9}$$

To further compare $d_m \ln(\theta_{h_m^*}^{\mathcal{C}_m}(\nu_m + \varepsilon))$ and $d\ln(\theta_{h^*}^{\tilde{\mathcal{T}}}(\varepsilon/2))$, we know that $\mathcal{C}_m$ is a subset of $\tilde{\mathcal{T}}$, thus $d_m \le d$. By combining Eq. (8), we can directly have

$$d_m \ln(\theta_{h_m^*}^{\mathcal{C}_m}(\nu_m + \varepsilon)) \le d\ln(\theta_{h^*}^{\tilde{\mathcal{T}}}(\varepsilon/2)) \,, \tag{10}$$

and

$$d_m \ln(\theta_{h_m^*}^{\mathcal{C}_m}(\nu_m + \varepsilon)) + \ln\left(\frac{\ln(1/\varepsilon)}{\delta}\right) < d\ln(\theta_{h^*}^{\tilde{\mathcal{T}}}(\varepsilon/2)) + \ln\left(\frac{\ln(2/\varepsilon)}{\delta}\right) \,. \tag{11}$$

Combining the above results, we have the following deductions, which lead to the conclusion.

$$\theta_{h_m^*}^{\mathcal{C}_m}(\nu_m + \varepsilon)\left(\frac{\nu_m^2}{\varepsilon^2} + \ln\left(\frac{1}{\varepsilon}\right)\right)\left(d_m \ln(\theta_{h_m^*}^{\mathcal{C}_m}(\nu_m + \varepsilon)) + \ln\left(\frac{\ln(1/\varepsilon)}{\delta}\right)\right)$$

$$\le \frac{\varepsilon + 2\nu_m}{\varepsilon + \nu_m}\theta_{h^*}^{\mathcal{C}_m}(\nu_m + \varepsilon)\left(\frac{\nu_m^2}{\varepsilon^2} + \ln\left(\frac{1}{\varepsilon}\right)\right)\left(d_m \ln(\theta_{h_m^*}^{\mathcal{C}_m}(\nu_m + \varepsilon)) + \ln\left(\frac{\ln(1/\varepsilon)}{\delta}\right)\right) \text{ \# by Eq. (5)}$$

$$\le \theta_{h^*}^{\mathcal{C}_m}(\frac{3}{2}\nu_m + \frac{3}{4}\varepsilon)\left(\frac{\nu_m^2}{\varepsilon^2} + \ln\left(\frac{1}{\varepsilon}\right)\right)\left(d_m \ln(\theta_{h^*}^{\mathcal{C}_m}(\frac{3}{2}\nu_m + \frac{3}{4}\varepsilon)) + \ln\left(\frac{\ln(1/\varepsilon)}{\delta}\right)\right) \text{ \# by Eq. (7)}$$

$$< \theta_{h^*}^{\mathcal{C}_m}(\varepsilon/2)\ln(2/\varepsilon)\left(d\ln(\theta_{h^*}^{\mathcal{C}_m}(\varepsilon/2)) + \ln\left(\frac{\ln(2/\varepsilon)}{\delta}\right)\right) \text{ \# by Eq. (8)(9)(11)}$$

$$\le \theta_{h^*}^{\tilde{\mathcal{T}}}(\varepsilon/2)\ln(2/\varepsilon)\left(d\ln(\theta_{h^*}^{\tilde{\mathcal{T}}}(\varepsilon/2)) + \ln\left(\frac{\ln(2/\varepsilon)}{\delta}\right)\right) \text{ \# by Lemma iii}$$

$\square$

# B    Experimental Details and Additional Results

## B.1    Empirical Settings

**Specifications of Multiple Target Models**    We take the following 12 specifications from a recent NAS work OFA [1] as our target models:

- s7edge_lat@88ms_top1@76.3_finetune@25
- s7edge_lat@58ms_top1@74.7_finetune@25
- s7edge_lat@41ms_top1@73.1_finetune@25
- s7edge_lat@29ms_top1@70.5_finetune@25
- note8_lat@65ms_top1@76.1_finetune@25
- note8_lat@49ms_top1@74.9_finetune@25
- note8_lat@31ms_top1@72.8_finetune@25
- note8_lat@22ms_top1@70.4_finetune@25
- note10_lat@22ms_top1@76.6_finetune@25
- note10_lat@16ms_top1@75.5_finetune@25
- note10_lat@11ms_top1@73.6_finetune@25
- note10_lat@8ms_top1@71.4_finetune@25

In the experiment with different number of target models (cf. Sec. 6.3), we empirically take the first $2, 4, 6, 8$ specifications from the above model configuration list as the target models set.

Table i: Win/Tie/Lose (W./T./L.) results of DIAM versus the other methods with varied numbers of queried batch based on paired $t$-tests at $0.05$ significance level. The comparisons are based on the performances of 12 target models after each query.

| Algorithms | Number of queried batch ($1,500$ examples per batch) | | | | | | | | | | W./T./L. |
|---|---|---|---|---|---|---|---|---|---|---|---|
| | 1 | 2 | 3 | 4 | 5 | 6 | 7 | 8 | 9 | 10 | |
| MNIST | | | | | | | | | | | |
| CAL | Win | Win | Tie | Win | Win | Win | Win | Tie | Win | Win | 8/2/0 |
| Entropy | Tie | Win | Tie | Win | Win | Win | Win | Tie | Win | Win | 7/3/0 |
| Margin | Win | Win | Tie | Win | Win | Win | Win | Tie | Win | Win | 8/2/0 |
| Least conf. | Tie | Win | Win | Win | Win | Win | Win | Tie | Win | Win | 8/2/0 |
| Coreset | Win | Win | Win | Win | Win | Win | Win | Tie | Win | Win | 9/1/0 |
| Random | Win | Win | Tie | Win | Win | Win | Win | Tie | Win | Win | 8/2/0 |
| W./T./L. | 4/2/0 | 6/0/0 | 2/4/0 | 6/0/0 | 6/0/0 | 6/0/0 | 6/0/0 | 0/6/0 | 6/0/0 | 6/0/0 | 48/12/0 |
| Kuzushiji-MNIST | | | | | | | | | | | |
| CAL | Win | Win | Win | Win | Win | Win | Win | Win | Win | Win | 10/0/0 |
| Entropy | Win | Win | Win | Win | Win | Win | Win | Win | Win | Win | 10/0/0 |
| Margin | Win | Win | Win | Win | Win | Win | Win | Win | Win | Win | 10/0/0 |
| Least conf. | Win | Win | Win | Win | Win | Win | Win | Win | Win | Win | 10/0/0 |
| Coreset | Win | Win | Win | Win | Win | Win | Win | Win | Win | Win | 10/0/0 |
| Random | Win | Win | Win | Win | Win | Win | Win | Win | Win | Win | 10/0/0 |
| W./T./L. | 6/0/0 | 6/0/0 | 6/0/0 | 6/0/0 | 6/0/0 | 6/0/0 | 6/0/0 | 6/0/0 | 6/0/0 | 6/0/0 | 60/0/0 |

## B.2 Computational Resources and Running Time

We run our experiments on 3 cloud servers, each of them has 128GB memory and 4 RTX 2080 graphic cards. The CPU is Intel Xeon Silver 4110 @ 2.10GHz with 8 cores. Since we run each of the compared method on one graphic card, respectively, we report the resource occupation of each individual process. The minimum requirement to train and validate the model is 10GB memory and 11GB CUDA memory with 128 batch size, respectively. If running the coreset query method, 10GB extra memory is needed to store the distance matrix.

For the running time, since there are multiple target models with varying complexities, they have different training and inference speed. The real time (calculated by gettimeofday() function) of sequentially training 12 target models on the initially labeled dataset, i.e., $3,000$ examples, is 00:26:42 (hh:mm:ss). For the data selection phase, i.e., select $1,500$ examples from $40,000$ unlabeled data, the real time of different methods are reported as following: random takes 1.73 seconds, least confidence takes 00:08:29 (need to evaluate the unlabeled data with each of the target model), margin takes 00:08:26, entropy takes 00:08:11, coreset takes 00:09:37. For the proposed DIAM method and CAL method, they need to evaluate the unlabeled data with the models trained with later epochs, which roughly take the size of the well-performed hypotheses set times of that of the entropy method to make the data selection.

## B.3 Additional Experimental Results

**Significance of Performance Comparison** We further report the significance results of the performance comparisons (cf. Figure 1 in the paper). Specifically, the win/tie/lose results of the performances of 12 target models after each query based on paired t-test at 0.05 significance level are reported in Table 1. The results show that our method can usually outperform the other compared methods significantly, which demonstrate that DIAM can improve all the target models simultaneously, but not paying too much attention to the specific models. This property is essential to the multiple target models applications. Because the target models are usually of equal importance, even though they have different prediction accuracies.

**Learning Curves of the Study on Different Numbers of Target Models** We plot the entire learning curves of the compared methods with different numbers of target models in Fig. i. It can be observed that the proposed DIAM method can surpass the traditional active and passive learning methods under different numbers of target models in most cases. These results reveal that our method

is robust to the number of target models, i.e., the data in the joint disagreement region is beneficial to all the target models.

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

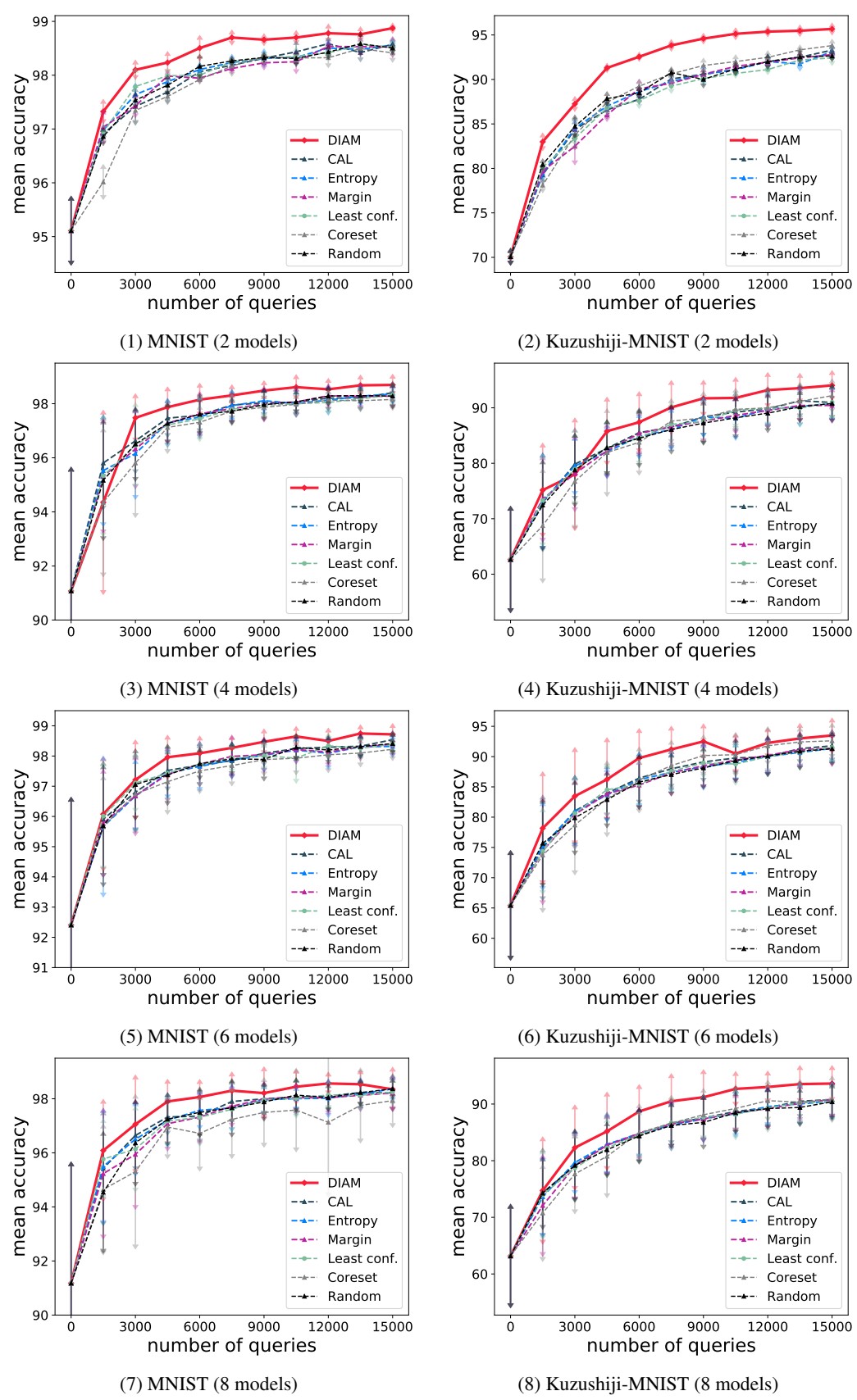

Figure i: Learning curves of the compared methods with different numbers of target models (2, 4, 6, 8 models). The error bars indicate the standard deviation of the performances of target models.