# OpenReview forum: "Active Learning for Multiple Target Models"
_NeurIPS.cc/2022/Conference — NeurIPS 2022 Accept_

### Official Review · Reviewer_FvK3 · 2022-07-06

**Rating:** 6
**Confidence:** 1
**Soundness:** 2 fair
**Presentation:** 2 fair
**Contribution:** 2 fair

**Summary:**

This paper proposes a novel problem setting for active learning, where the aim is to query samples to improve multiple target models' performance simultaneously. This paper theoretically analyzes the label complexity of active learning under this setting and shows that active learning has the potential to achieve better label complexity. In addition, based on this analysis, this paper proposes an active learning strategy to query samples in the region of the joint disagreement of different target models, and experimentally shows the effectiveness of this strategy with two OCR datasets.

**Questions:**

* it is better to explicitly define the terms "realizable and agnostic cases" for improving clarity and readability.
* In Theorem 1, each classifier $\hat{h} _i$ is obtained by ${\rm min} _{C_i} \ d(h_i, h_A)$. However, can such $( \hat{h} _i ) _i$ be learned from the same data? This point is unclear to me.
* In Line 230, it would be better to explain the hyperparameter $q$.

**Limitations:**

The paper addressed the limitations in page 6 but does not mention any potential negative social impact.

**Strengths And Weaknesses:**

### Strengths
* The problem setting tackled in this paper seems to be interesting and practically useful although it is difficult to judge novelty correctly for me because I'm not an expert in this field.
* There is a theoretical analysis. It seems intuitive and reasonable to select samples in the disagreement area of multiple target models. However, since I'm not an expert in the field, I was not able to follow all the proof and statements.
* Experimental results show the effectiveness of this method.

### Weaknesses
* Since the proposed method seems to be a general algorithm, it might be better to include evaluation other than OCR benchmark datasets.

---

> ### Author Response · Authors · 2022-08-01
> **We appreciate the reviewer for providing valuable feedback and comments to our manuscript. We address the reviewer's concerns as below.**
>
> > Q: it might be better to include evaluation other than OCR benchmark datasets.
>
> A: Thanks for the comment. We motivate the multi-model setting from the case of developing machine learning systems for diverse devices. OCR is one of the representative task with such requirement. Thus, we primarily validate our method on these tasks, in which, Kuzushiji-MNIST dataset has 70,000 data points. More practical machine learning tasks will also be considered in our future work.
>
> > Q:It is better to explicitly define the terms "realizable and agnostic cases" for improving clarity and readability.
>
> A: Thanks for the advice, we have made this clearer in the revised paper, e.g., in L133, L196 in the revised paper.
>
> > Q: In Theorem 1, each classifier $\hat h_i$ is obtained by $\min_{{h_i}\in \mathcal{C}_i}d(h_i, h_A)$. However, can such $\hat h_i$ be learned from the same data? This point is unclear to me.
>
> A: This is a good question. Unfortunately, learning a good model from different hypothesis classes may require very different data. On the contrary, finding a closest classifier for a given hypothesis usually can be done by existing data. One technique that accord with this is the knowledge distillation, which tries to shape the prediction of a small model to be similar with a larger model on existing data. We have added discussion on this in the revised version.
>
> > Q: In Line 230, it would be better to explain the hyperparameter q.
>
> A: Thanks for the suggestion. We have added more explanation about this hyperparameter q in the Sec. 5 in the revised paper. More concretely, we state that _The hyperparameter $q$ controls the conservativeness of the algorithm. With a larger $q$, it will reject more less informative unlabeled data in the online setting. When $q=1$, the algorithm degenerates to query the data point which falls into any disagreement regions._
>
> > Q: The paper does not mention any potential negative social impact.
>
> A: Thanks for pointing out this problem. We have discussed about this part in the revised paper, e.g., in the last paragraph in Sec. 5.

---

> > ### Comment · Reviewer_FvK3 · 2022-08-08
> > **Thank you for the response**
> >
> > Thank you for your response.  I want to keep my initial rating.

---

### Official Review · Reviewer_mjax · 2022-07-07

**Rating:** 5
**Confidence:** 3
**Soundness:** 4 excellent
**Presentation:** 2 fair
**Contribution:** 3 good

**Summary:**

This paper studies an important setting of active learning, motivated by the need to efficiently acquire a single dataset that is useful to simultaneously train various models/architectures. Contrary to common observations against a possible improvement of AL over PL under this setting, the authors theoretically prove a label complexity bound of active learning for multiple models which combined with a bound on passive learning implies active learning may also be efficient under this setting. The authors then modify a disagreement-based algorithm used for single-model AL to that for multiple models by taking their average disagreement and prove its label complexity in the agnostic setting. Lastly, this paper proposes a heuristic to circumvent the computational complexity when employing neural networks as the hypothesis class, and empirically demonstrates that this method outperforms well-known active learning algorithms designed for single-model AL, extended using the same average over multiple models, under this multiple target setting.

**Questions:**

Could the authors elaborate on the remark after Theorem 2? The fact that the target concept h* may not be included in every hypothesis space is not unique to PL, and is problematic for AL as well, no? Further, why is the last sentence in Section 4.1 included here? It seems that this would better fit under Section 4.2.

Please correct me if there are any misunderstandings in my comments above.


**Limitations:**

This draft does not clearly state apparent limitations, but does mention future work. Please incorporate my comments on the algorithmic and computational limitations and add others as appropriate.

**Strengths And Weaknesses:**

The main (and only) major theoretical contribution in this paper seems to be Theorem 1: It is an improved analysis method over a straightforward proof technique that translates the sample complexity of AL applied to a single hypothesis class indexed by i to that for multiple hypothesis classes as the sum of each complexity, as the authors mention (line 125). For PL, one can easily obtain a stronger translation since by definition of passive learning (random sampling), the complexity of PL for multiple hypothesis classes is the maximum over each sample complexity. I would think that for PL, this is the only guarantee possible without additional assumptions on how the hypothesis classes are related. Theorem 1 is a tool that is used to get a stronger translation from AL bounds for a single model to that for multiple models, and the authors use this to show that AL can potentially achieve an exponential improvement over PL in the multiple-model setting.

This is an interesting conclusion, but the reason I say this is the only theoretical contribution is that these bounds and corresponding comparisons between AL and PL are well-known for the single-model case, which this work uses extensively. In proving the theoretical statements in the paper, the only non-trivial modification to proofs by Henneke under this multiple-model setting is to use Theorem 1 in place of the sum of sample complexities for single-model AL.  This isn’t necessarily bad, but I would appreciate if the authors mention whether it’s possible that a stronger translation bound in place of Theorem 1 is likely. Additionally, I would appreciate if the authors remark on whether the bound translation for PL (as I mentioned above) is perhaps the tightest possible, and if not, it would be nice to see an effort improving the PL bound for multiple models being max_i \Lambda_i to another method analogous to Theorem 1 for AL.

Despite these concerns on theoretical statements, it is nice to see Theorem 4 which shows that DIAM-online is better than CAL applied to multiple target models. However, its presentation could be improved by instead writing that the sample complexity of DIAM-online (e.g. by writing \Lambda_m (DIAM) < \Lambda_m (CAL)), as the terms are redundant.

Here are a few major comments
* Presentation could be improved significantly. Many statements are defined in-text much after they are used, and some notations are never defined formally. Word choices are also poor, making it hard to infer what the authors attempt to say. There are some redundant statements that make the reader confused whether the statements are regarding different situations, and assumed settings go back and forth between agnostic and realizable. How the disagreement regions are to be computed in practice is described briefly in the last paragraph, whereas this the algorithm used throughout experiments.
* I like how despite this paper being mostly theoretical, the experiments are done using somewhat large models and resembles the motivation mentioned in the introduction. However, I don’t think the tasks being classification on MNIST is convincing and I would like to see experiments on at least one larger dataset (even CIFAR-10 would be fine, or others used in the base code by Cai et al. 2020).
* The proposed problem setting is interesting and well-motivated. This work is the first to theoretically analyze the sample complexity of active learning for multiple target models, when the hypothesis classes are fixed (due to different target deployments). Section 2 compares with related work that studies similar but different problems and helps understand this problem setting’s position within literature.
* Algorithmically, the downside is that DIAM probably doesn’t work in the batch-setting, i.e. the algorithm would collect many similar samples when present. This is of large interest in the single-model AL setting, and I think the authors should mention this in the paper as a limitation.
* Also, the computational complexity should also be mentioned. As the authors note, computing the disagreements is computationally prohibitive for deep models. It would be nice to see how the heuristic overcomes this difficulty and what the resulting runtime is for practically-sized neural networks.

Minor comments
* Many grammatical errors should be fixed, e.g. remove “with” in line 83, must has -> is in line 125 and move the parenthesis before the period or even better, remove the parenthesis brackets and fix accordingly.
* Notations could be improved, but this might require much effort. For example, \Lambda_i is not defined other than the description in parenthesis in line 125-126, although I understand it to be the single-model AL complexity for class C_i (explained later in the draft). The fact that \Lambda_m is the notation for multiple-model AL where m is a different meaning than the “i” in \Lambda_i makes it confusing at first read.
* I don’t think the statement in Theorem 2 should be stated as a “Theorem”. I recommend changing this to a proposition or writing it as an in-text sentence after line 125, since the two statements are related in how to obtain a multiple-model sample complexity for AL (line 125) and PL (this statement).

In summary, two major concerns I have is originality and presentation. For originality, I would like to hear the authors' remarks on whether Theorem 1 is tight and how a bound on single-model PL may or may not be improved.

---

> ### Author Response · Authors · 2022-08-01
> **We sincerely thank the reviewer for the time and efforts, as well as the constructive comments. We have carefully revised the paper accordingly. In the following, we first address the major concerns raised by the reviewer, then the responses to the other questions are presented latter. Please also see the revised paper (has been uploaded in OpenReview, we have highlighted the updated contents).**
>
> > Whether it’s possible that a stronger translation bound in place of Theorem 1 is likely.
>
> Thank for this insightful question. We believe there does exist improvement of Theorem 1, since it does not make additional assumptions on how the hypothesis classes are related or satisfy certain conditions. Improvement should be obtained in more specific settings.
>
> For the general bound, we are still looking for a better solution yet. Specifically, we are currently working on the shattering technique, which exploits the probability of a data points that can be shattered by the current version space V (cf. [14]). We will report our latest findings on arXiv once we have noticeable progress, thanks.
>
> > Whether the bound translation for PL is perhaps the tightest possible.
>
> This is a very interesting question. The bound translation for PL is actually quite tight. However, it depends on the property of the given model (e.g., $\nu$), as stated in the L181 in the paper. On the other side, we note that Theorem 1 can also be applied to PL. Considering that the target concept falls into one of the hypothesis space j which is far away from another one k, in this case, we may expect that finding a $\epsilon/2$-good classifier from j (cf. Theorem 1) requires less data than finding an $\epsilon$-good classifier from k (by comparing Eq. (5) and (6)).
>
> Nevertheless, we emphasis that this does not mean that AL can hardly surpass PL in the multiple models setting. Because many AL methods are guaranteed to have better label complexity than PL, e.g., those analyzed by Hanneke, which means they need less labels to obtain a $\epsilon/2$-good classifier from the combined hypothesis space. Thus, although Theorem 1 works for both AL and PL, AL is still with more potential.
>
> > Presentation could be improved significantly.
>
> Thanks for the comment. We have carefully proofread the paper, and optimized the presentations. Specifically, we define and change some notations for clarity, for example, the label complexity of multiple models $\Lambda_m$ has been replaced by $\Lambda^M$. The symbols $D_X, D_y$, $\nu$ are formally defined in Sec. 3.1. To improve the paper structure, Theorem 2 is removed and the label complexity of PL for multiple models is presented as in-text in L130 in the revised paper. We rearrange Sec. 3 and Sec. 4 according to the comments to improve the readability. We also remove the redundant terms in Theorem 3, and augment the introduction of the implementation of DIAM for the deep models at the end of Sec. 5.
> Besides, we rectify many grammatical errors, which include removing the redundant prepositions (e.g., tackle with), moving the parenthesis before the period, an hypothesis -> a hypothesis, etc.
>
> > I would like to see experiments on at least one larger dataset.
>
> Thanks for the comment. We motivate the multi-model setting from the case of developing machine learning systems for diverse devices. OCR is one of the representative task with such requirement. Thus, we primarily validate our method on these tasks, in which, Kuzushiji-MNIST dataset has 70,000 data points. More practical machine learning tasks will also be considered in our future work.
>
> > DIAM probably doesn’t work in the batch-setting, the authors should mention this limitation.
>
> Thanks for the comment. We have also pointed out this limitation and stated the future plan in the revised paper.
> In this work, we define the problem setting of AL for multiple models and propose an informativeness-based criterion for the AL under multiple models setting, the study on representativeness is scheduled as a future work. More concretely, we are going to exploit the multiple feature representations of the data (obtained with different model architectures), which may relate to the multi-view clustering. For the current version of the method, we heuristically sample a subset from the top-rated unlabeled data to introduce the diversity for batch-mode data selection (as stated in L269 in the original paper).
>
> > The computational complexity should also be mentioned.
>
> The running time of different compared methods are reported in the supplementary materials (cf. Sec. B.2), which show that our proposed heuristic is efficient since it only requires predict the unlabeled data at each of the later epochs. We have also discussed it at the end of Sec. 5 in revised paper, thanks.
>
> > In Theorem 2, the fact that the target concept h* may not be included in every hypothesis may be problematic for AL as well.
>
> Yes, the same problem is also confronted by AL. However, according to Theorem 1, we do not care about the label complexity for a specific hypothesis space, but try to find a good classifier from the combined hypothesis space, thus it is not problematic for the AL.
>
> > why is the last sentence in Section 4.1 included here? It seems that this would better fit under Section 4.2.
>
> We have rectified this by rearranging Sec. 4 in the revised paper, thanks for the advice.

---

> > ### Comment · Reviewer_mjax · 2022-08-04
> > **Response**
> >
> > Thank you for the detailed response. The authors appropriately addressed my questions and I updated my rating from 4 to 5 accordingly. I believe that larger-scale experiments would strengthen the algorithmic side of this paper, although this is not a main concern and merely a suggestion. The reason I didn't update my rating further is that I still think the presentation can be improved much more although it appears that this would take quite some effort.

---

### Official Review · Reviewer_xpsR · 2022-07-11

**Rating:** 8
**Confidence:** 4
**Soundness:** 3 good
**Presentation:** 3 good
**Contribution:** 4 excellent

**Summary:**

This work investigates an interesting problem, i.e., how to actively select the informative data to improve multiple target models simultaneously. First of all, the authors formally define the active learning (AL) for multiple target models problem and bridge the label complexity between AL for single and multiple models. After revealing the potential improvement of AL for this setting, a novel query strategy DIAM for multiple target models is further proposed, which prefers the data located in the joint disagreement regions. An efficient implementation for deep models is also provided. Both theoretical and empirical studies are conducted to validate the proposed approach.

**Questions:**

1. Theorem 1 also works for passive learning. How to guarantee the superiority of active learning?

2. Intuitively, the target models should share some specifications to make the active selection feasible. However, the proposed method works fine with very different target models according to the results in supplementary materials. The authors may explain more about this phenomenon.


**Ethics Review Area:**

["I don’t know"]

**Limitations:**

Yes

**Strengths And Weaknesses:**

This work is of high quality in my view due to the following reasons:

1.	The problem explored by this study is very important and meaningful. In many machine learning tasks, the devices to be deployed usually have varying computational resources, and there is a heavy burden of data labeling to train multiple models. These challenges become more tough with the flourishing applications of deep learning in recent years.
2.	From the technical perspective, active learning is usually believed to be model-dependent. This work presents a nice attempt towards multiple target models from both theoretical and empirical views. It broadens the learning scenarios of active learning and may inspire future work towards thinking of this problem. This contribution is significant in my opinion.
3.	The paper is well-structured and easy to follow. The authors also provide enough supplementary materials and code for reproduction.

Weaknesses：
1. It seems that the specific form of \sigma in DIAM method is not given in the paper, the authors only say that it has the same form with RobustCAL method. I suggest to add this part for a more self-contained presentation.
2. It is better to introduce the structure of target models in the paper rather than in the supplementary materials for better understanding the results.

---

> ### Author Response · Authors · 2022-08-01
> **We thank the reviewer for acknowledging the novelty of our work and appreciating its practical importance. Here, we address the reviewer’s main question.**
>
> > Q: It seems that the specific form of $\sigma$ in DIAM method is not given in the paper, the authors only say that it has the same form with RobustCAL method. I suggest to add this part for a more self-contained presentation.
>
> A: Thanks for the suggestion. We have formally defined $\sigma$ in the revised paper.
>
> > Q: It is better to introduce the structure of target models in the paper rather than in the supplementary materials for better understanding the results.
>
> A: Thanks for the suggestion. We have introduced more details of the target models in the revised paper.
>
> > Q: Theorem 1 also works for passive learning. How to guarantee the superiority of active learning?
>
> A: Many AL methods are guaranteed to have better label complexity than passive learning, e.g., CAL, which means they need less labels to obtain a $\epsilon/2$-good classifier from the combined hypothesis space. Thus, although Theorem 1 works for both AL and passive learning, AL is still with more potential.
>
> > Q: Intuitively, the target models should share some specifications to make the active selection feasible. However, the proposed method works fine with very different target models according to the results in supplementary materials. The authors may explain more about this phenomenon.
>
> A: Thanks for the insightful question. We speculate that the proposed DIAM method queries the data falls into the joint disagreement region, which guarantees the utility to each of the target models. Thus, even the target models have very different architectures, the average accuracies of them can be improved effectively. This phenomenon also reveals that DIAM has the potential to tackle more challenging situations, i.e., improving heterogeneous target models effectively.

---

### Official Review · Reviewer_rdsY · 2022-07-11

**Rating:** 6
**Confidence:** 3
**Soundness:** 3 good
**Presentation:** 3 good
**Contribution:** 3 good

**Summary:**

It is generally believed active learning is model-dependent. This paper studies the problem of active learning for multiple target models. The authors first analyze the label complexity for active learning under the setting with multiple target models, then find the label complexity of single model is an upper bound of that for multiple models under the realizable case. They also propose an agnostic disagreement-based selection criterion for the agnostic case. Experiments verify the effectiveness of the proposed approach.

**Questions:**

Theorem 3 is based on hyperparameter q=1. Is there a tighter result when q>1?

**Strengths And Weaknesses:**

This paper explores a new setting of active learning where there are multiple target models to be learned. How to design effective active learning algorithms is practical and important to the machine learning community. It first analyzes the label complexity of passive learning and active learning for multiple target models under realizable case, and advocates active learning has potential improvement under this setting. For the agnostic setting, the authors propose disagreement-based selection criterion to query examples located in the joint disagreement regions. Theoretical analysis shows the proposed method achieves better label complexity than that of CAL (a representative approach) under some ideal situations. The writing is good and clear. The theoretical analysis and empirical experiments are solid.

---

> ### Author Response · Authors · 2022-08-01
> **We thank the reviewer’s valuable comments and answer specific questions below.**
>
> > Q: Theorem 3 is based on hyperparameter q=1. Is there a tighter result when q>1?
>
> A: We are still working on a tighter result of q>1. Since this setting requires the discussion for the connection between the hypothesis classes and the data distributions, more effort is needed to obtain such result. We will continuously work on improving this theorem, thanks.

---

### Meta-Review · Area_Chair_4heB · 2022-08-25

**Recommendation:** Accept
**Confidence:** Certain

**Metareview:**

This paper studies a novel active learning setting adapted to learning multiple target model. The authors propose a setting that can benefit to all tasks by focusing on regions with high disagreements. This contribution shows in a sense that the active learning procedure can be transferable to multiple tasks. A theoretical analysis is provided in the form of a bound on label complexity. Experimental results support the claims.

The reviewers have globally appreciated the contribution and most the comments raised in the reviews have been addressed in the rebuttal.
The overall evaluation of the paper is positive and I propose acceptance.

I recommend the authors to take into consideration the last comments of the reviewers, in particular for improving the presentation of the paper.

**Award:**

No

---

### Decision · Program_Chairs · 2022-09-14

Accept